# Analysis of Thermal Degradation Kinetics of POM under Inert and Oxidizing Atmospheres and Combustion Characteristics with Flame Retardant Effects

**DOI:** 10.3390/polym15102286

**Published:** 2023-05-12

**Authors:** Dan Zhang, Siyuan Zhou, Mi Li

**Affiliations:** 1School of Chemistry and Chemical Engineering, Nanjing University of Science and Technology, Nanjing 210094, China; danzhang@njust.edu.cn (D.Z.); 15869114757@163.com (S.Z.); 2School of Mechanical Engineering, Nanjing University of Science and Technology, Nanjing 210094, China

**Keywords:** polyoxymethylene (POM), thermal decomposition, thermal kinetics, degradation mechanism, combustion performance

## Abstract

Degradation behavior of combustible fuel is the core factor in determining combustion characteristics. To investigate the effect of ambient atmosphere on the pyrolysis process of polyoxymethylene (POM), the pyrolysis mechanism of POM was studied with thermogravimetric analyzer tests and Fourier transform infrared spectroscopy tests. The activation energy, reaction model, and estimated lifetime of POM pyrolysis under different kinds of ambient gases have been estimated in this paper based on different results of the kinetics. The activation energy values, obtained with different methods, were 151.0–156.6 kJ mol^−1^ in nitrogen and 80.9–127.3 kJ mol^−1^ in air. Then, based on the Criado analysis, the pyrolysis reaction models of POM in nitrogen were found to be mastered by the “*n* + *m* = 2; *n* = 1.5” model, and by the “A3” model in air. The optimum processing temperature for POM was estimated, with a range from 250 to 300 °C in nitrogen and from 200 to 250 °C in air. IR analysis revealed that the significant difference in POM decomposition between N_2_ and O_2_ atmospheres is the formation of isocyanate group or carbon dioxide. Combustion parameters of two POMs (with and without flame retardants) obtained using cone calorimetry revealed that flame retardants can effectively improve the ignition time, smoke release rate, and other parameters of POM. The outcomes of this study will contribute to the design, storage, and transportation of polyoxymethylene.

## 1. Introduction

Polyoxymethylene (POM), structural formula HO(CH_2_O)_n_H, is the smallest polymerization product of formaldehyde, with a typical degree of polymerization of 8–100 units. Due to the fact that POM has the characteristics of high strength, wear resistance, fatigue resistance, good dimensional stability, self-lubrication, etc., it has been applied increasingly in mechanical engineering, the automotive industry, precision instruments, etc., either alone or as a composite material. For example, it is used to make parts for telephones, tape recorders, and computers, as well as shafts and gears [1,2,3]. Its role has become more important over the years due to its excellent physical and mechanical properties [4]. However, the unstable hemiacetal end groups on both sides of the molecular chain begin to break down when POM decomposes into formaldehyde gas once heated above 100 °C [5]. Therefore, in practical applications POM must be treated with appropriate processing. The prerequisite for processing is the knowledge of its pyrolytic behavior, and the study of pyrolysis of combustible solids is the basis for its combustion. Therefore, the study of the pyrolysis process of POM can provide recommendations for its processing as well as for further studies of its combustion properties. Studies on pyrolysis of POM have appeared in a number of works. Most of these studies have been developed with the calculation of kinetic parameters. The thermal degradation profiles of POM/thermoplastic polyurethane (TPU) blends were investigated using TG and FTIR by Pielichowski et al. It was found that the incorporation of TPU into the POM matrix improved the thermal stability of the blends compared to the original material, and it was found that the thermal degradation of POM is basically a one-step process [6]. Fayolle et al. investigated the thermal oxidation of unstable POM using gravimetric and infrared spectroscopic methods at 90, 110, and 130 °C and at various oxygen pressures from 0 to 2.0 Mpa [7]. Studies on reaction models of POM had been relatively sparse until Lüftl S et al. confirmed the applicability of the Flynn-Wall method, and thus determined the kinetics of the pyrolysis reaction of POM [8].

But identification of reaction models using the Coats–Redfern and Criado methods has not been performed. In addition, few researchers have studied the estimated lifetime and degradation mechanism of POM under different atmospheres. In this paper, the pyrolysis process and kinetic analysis of POM are investigated with thermogravimetric tests and Fourier transform infrared spectroscopy (FTIR) tests under nitrogen and air.

Combustion is the next step in the thermal decomposition of the material, and some research has been carried out on the combustion properties of POM. Shaklein et al. explored the effect of reactor geometry on polymer combustion [9]. Korobeinichev et al. investigated the thermal decomposition and combustion of horizontally placed POM sheets, and determined the kinetic parameters of thermal degradation of POM assuming two parallel reactions [10]. By using temperature profiles measured in the gas phase with the micro thermocouple technique, the combustion surface temperature and mass burning rate of POM were determined by Glaznev et al. [11]. It is necessary to improve the combustion performance of POM in order to make it better for use; flame retardants are often used in the field of materials to improve the combustion properties of materials. Commonly used flame retardants are boron and molybdenum system, nitrogen system, and phosphorus–nitrogen mixed flame retardants [12,13,14]. Phosphorus–nitrogen flame retardant is a compound of phosphorus and amine, and the nitrogen oxide released during combustion can play the role of flame suppression, as well as prevent contact between oxygen and carbon, which is more suitable for POM; therefore, phosphorus–nitrogen flame retardant was chosen in this study.

Research on the application of flame retardants in the combustion behavior of POM is still relatively sparse, so this paper selects phosphorus–nitrogen flame retardant to investigate the combustion performance of two kinds of polyacetal materials (with and without flame retardants added) to supplement some experimental data for research in the field of polyacetal. This paper can provide references and suggestions in terms of pyrolysis reaction model, lifetime estimation, combustion behavior of POM, and the effect of flame retardants.

## 2. Experimental Procedure

The POM ingredient (density: 1.43 g cm^−3^, purity: ≥99%) used was a commercial product of high purity manufactured by the DuPont Co. of Wilmington, DE, USA. TG tests were obtained by using a SDT-Q600 thermal analyzer (TA Co., New Castle, DE, USA) with a microbalance sensitivity of 0.1 μg and a temperature sensitivity of 0.001 °C. The sample mass of each experiment was 2 mg and the initial mass error of the sample did not exceed 0.1 mg. Conventional constant heating rate TG measurements were run at 5, 10, 15, and 20 °C min^−1^. The flowing gas was N_2_ and air at a flowing rate of 100 mL min^−1^. The FTIR technique (PerkinElmer TL8000, Waltham, MA, USA) with a purge flow of 35 mL min^−1^ nitrogen or air at 20 °C min^−1^ heating rate was employed to analyze the evolved gas. The combustion performance of two different types of POM was studied by using a conical calorimeter model 0007 from FTT (Fire Testing Technology), UK, with three settings of thermal radiation intensity (25 kW/m^2^, 35 kW/m^2^, and 50 kW/m^2^).

## 3. Theoretical Consideration

The kinetics of polymer degradation are described by Equation (1) [15], where α is the level of conversion, *T* means the environmental temperature, *β* means the heating rate, *A* the pre-exponential factor, *E_a_* the activation energy, *f* (α) the different function of conversion, and *R* the gas constant. The conversion, *α*, is calculated in terms of weight loss by Equation (2) [16], where *W*_0_ is the initial weight of the sample, *W_t_* is the weight of the sample at time *t*, and *W_f_* is the weight of the completely decomposed sample. The algebraic expression for kinetic models commonly used [17] are exhibited in Table 1.
(1)$$\beta \frac{d\alpha }{dT}=Af\left(\alpha \right)\exp\left(-\frac{E_{a}}{RT}\right)$$
(2)$$\alpha =\frac{W_{0}-W_{t}}{W_{0}-W_{f}}$$

Without any assumption on the decomposition model, the isoconversional methods (Friedman (FR), Flynn–Wall–Ozawa (FWO), and Kissinger–Akahira–Sunose (KAS) methods) can give activation energy *E* as a function of conversion by using different heating rates. The Friedman method [18] utilizes the TG data of different heating rates to calculate *E_a_*. As the mathematical plot of ln(d*α*/d*t*) against 1/*T* shows in Equation (3), a linear relationship could be obtained with a slope equal to −*E_a_*/R. The FWO method [19] is an integral method, which is independent of the degradation mechanism. The logarithmic form can be given as Equation (4). The activation energy can be obtained from plot of ln*β* versus 1/*T* at a fixed conversion with the slope being 0.4567 *E_a_*/RT. Analogous to the FWO method, KAS [20] is an integral calculation method and the equation of it can be given as Equation (5). Plotting ln(*β*/*T^2^*) against 1/*T* allows *E_a_* to be calculated for each degree of conversion value.
(3)$$\ln\frac{d\alpha }{dt}=\ln\left[Af\left(\alpha \right)\right]-\frac{E_{a}}{RT}$$
(4)$$\mathrm{lg}\beta =\ln\frac{AE_{a}}{Rg\left(\alpha \right)}-2.315-0.4567\frac{E_{a}}{RT}$$
(5)$$\ln\frac{\beta }{T^{2}}=\ln\frac{AR}{E_{a}g\left(\alpha \right)}-\frac{E_{a}}{RT}$$

The Criado model is applied to further confirm reaction mechanism through comparing the fitting degree of curves. By combining Equations (1), (5) and (6) is obtained [21], where 0.5 refers to the conversion of 0.5. The left side of Equation (6), Z(*α*)/Z(0.5), is a reduced theoretical curve which is characteristic of each reaction mechanism, whereas the right side of Equation (6) is associated with reduced rate. Through comparing the coincidence degree of plots, we can conclude that the kinetic model describes an experimental reactive process. Once a reaction mechanism is determined, the reaction order (*n*) could be quantitative, which is essential to predicting thermal lifetime.
(6)$$\frac{Z\left(\alpha \right)}{Z\left(0.5\right)}=\frac{f\left(\alpha \right)g\left(\alpha \right)}{f\left(0.5\right)g\left(0.5\right)}={\left(\frac{T_{\alpha }}{T_{0.5}}\right)}^{2}\frac{{\left(d\alpha /dt\right)}_{\alpha }}{{\left(d\alpha /dt\right)}_{0.5}}$$

Then, based on a single heating rate measurement, the Chang method is used to evaluate the activation energy *E_a_* and the frequency factor *A* without making any assumptions. The Chang method [22] is a one-heating-rate treatment method. As shown in Equation (7), a plot of ln[(d*α*/d*t*)/(1 − *α*)*^n^*] versus 1/*T* yields a straight line if the decomposition order is right. The slope and intercept of this line can provide the *E_a_* and ln*A* values, respectively.
(7)$$\ln\left[\frac{\left(d\alpha /dt\right)}{{\left(1-\alpha \right)}^{n}}\right]=\ln A-\frac{E_{a}}{RT}$$

After the three necessary parameters (*n*, *E_a_,* and *A*) are obtained, the half-life time *t*_1/2_ and estimated lifetime *t*_f_ can be predicted eventually. The *t*_f_ and estimated *t*_1/2_ are defined to be the time when weight losses reach 0.5 and 0.05 as shown in Equations (8) and (9), respectively [23].
(8)$$t_{f}=\frac{\left(0.95^{1-n}-1\right)}{Z\left(n-1\right)}\exp\left(\frac{E}{RT}\right)\left(n\neq 1\right)$$
(9)$$t_{1/2}=\frac{\left(0.5^{1-n}-1\right)}{Z\left(n-1\right)}\exp\left(\frac{E_{a}}{RT}\right)\left(n\neq 1\right)$$

Data such as ignition time, heat release rate, total heat release, smoke release rate, and total smoke release can be obtained using a cone calorimeter, and repeatability of operation can also be achieved [24]. Time to ignition (TTI) is one of the main parameters used to characterize the fire hazard of a material. TTI reflects the ease with which a material can be ignited. A longer ignition time indicates that it is harder for the material to be ignited, and that it is less of a fire hazard and has better resistance to fire [25]. The heat release rate (HRR) represents the rate of heat release per unit area after ignition of a sample under a certain heat flow intensity. HRR can be further subdivided into peak rate (PkHRR) and mean rate (MHRR) [26]. The total heat release refers to the heat released per unit area of the material combustion process [27], and the total heat release is almost independent of the value of other external factors when ventilation is adequate, so the total heat release is usually used as one of the parameters to evaluate the fire hazard of a material. Smoke release rate and total smoke release can reflect the degree of pollution of the material to the environment, and can also reflect the degree of combustion of the material [28].

## 4. Results and Discussion

### 4.1. Thermal Degradation Behaviour

The TG–DTG curves of POM in nitrogen and air are shown in Figure 1. The trend of the curves of mass loss is almost the same, and the curve of DTG data exhibits only one degradation peak. It can be speculated that the effects of different atmospheres on the pyrolysis of POM are relatively similar. This is due to the same chemical bond molecular structure. Consulting the data listed in Table 2, it can be seen that the onset degradation temperature (*T*_onset_), the end temperature (*T*_end_), and the peak temperature (*T*_p_) in air are reduced by 60–120 °C compared to those in nitrogen, indicating that POM in air begins degradation at lower temperature. Results imply that the thermostability of POM is a little better in nitrogen than in air. Various arguments have been proposed by researchers to explain the mechanism of atmosphere effects on thermal behavior for polymers [29,30]. One of the popular explanations is that a change in reaction mechanism occurs when there is a change in atmosphere [31].

### 4.2. Kinetics Calculation with Isoconversional Kinetic Methods

To calculate activation energy, four heating rates (5, 10, 15, and 20 °C min^−1^) and three isoconversional kinetic methods (the FR, FWO, and KAS methods) are applied here. According to Equations (3)–(5), plots of ln(*dα*/*dT*) against 1/*T*, log (*β*) against 1/*T*, and ln(*β/T^2^*) against 1/*T* are used, respectively, as shown in Figure 2. According to the linear relationship between the different variables, the corresponding activation energy (*E_α_*) is obtained at each conversion.

As shown in Figure 3, *E_a_* for POM in nitrogen is 151.0–156.6 kJ mol^−1^, while that in air is 80.9–127.3 kJ mol^−1^ (varying with the kinetic methods). The derived *E_a_* in air is much less than that in inert gas, reflecting that the purge gas has a great effect on reactivity for POM. Therefore, compared to an inert atmosphere, the POM requires lower temperature and less energy to decompose in an oxidizing atmosphere. Generally speaking, *E_a_* values are continuous from α = 0.1 to 0.9, showing that there is no change in the reaction mechanism. However, *E_a_* values obtained using the Friedman method are obviously smaller than the other two methods, which is mainly due to the conservative property of the Friedman method [32].

### 4.3. Degradation Mechanism Determination

Figure 4 compares the theoretical master curves to the experimental curves obtained from TG curves at 10 °C min^−1^ under two kinds of atmospheres. The experimental curves for POM degradation in nitrogen and in air are almost totally covered by the master curves of Z (*n* + *m* = 2; *n* = 1.5)/Z (0.5) and Z(A3)/Z (0.5), respectively. Once confirming the kinetic models for POM degradation in nitrogen and in air, the decomposition order (n) for POM degradation then can be determined to be 3/2 in nitrogen and 1/3 in air.

### 4.4. Kinetics Calculation with the Chang Model

Figure 5 shows the relationship given by Equation (7) of the Chang method. Activation energy (*E*_a_) and frequency factor (*A*) for POM at 10 °C min^−1^ heating rate can be quantitatively calculated, as summarized in Table 3. From the above calculation with the Friedman, FWO, and KAS methods, it can be indicated that kinetic parameters change more or less with the temperature. Despite this, it is believed that this variation is less in the Chang method, as Table 3 shows that all correlation coefficients R^2^ for POM are above 0.98, indicating good linearity through a wide range of temperatures. *E*_a_ for POM in nitrogen is 276.1 kJ mol^−1^, while that in air is 244.3 kJ mol^−1^, representing that degradation reactions under nitrogen are more difficult to process. In addition, when the value of degradation reaction order (*n*) is higher, degradation reaction occurs more slowly, reflecting higher thermal stability for POM in nitrogen than in air. Unlike the model-free method, which reveals complexity of the process in the form of a functional dependence of *E*_a_ on *α*, the Chang method has the ability to yield a single effective value of *E*_a_ for the whole process. Although the *E*_a_ value using the Chang method is different from that using model-free methods, the Chang method is capable of reasonably predicting estimated lifetime for polymers.

### 4.5. Thermal Lifetime Prediction

Calculating the thermal kinetic parameters can be used to estimate the maximum useable temperature, the optimum processing temperature regions, and the estimated lifetime of polymer materials [33]. The decomposition kinetics at high temperatures can predict the lifetime under service conditions [34]. The important lifetime parameters for POM at different temperatures in nitrogen and air have been calculated, as summarized in Table 4, reflecting that the lifetime parameters decrease progressively with the increasing temperature. The lifetime of the POM under 250 °C in nitrogen could reach 18 days, but this would be sharply reduced to 3 h in air. The static processing time for POM at 350 °C will last for 0.944 min in nitrogen but for 0.002 min in air. The evaluated values suggest the optimum processing temperature range to be between 250 to 300 °C in nitrogen and between 200 to 250 °C in air. The lifetime parameters would provide a simple approach for quality-control experiments by using an accelerating aging process.

### 4.6. Evolved Gas Analysis

The 3D and 2D infrared spectrum of evolved gas, and identified IR frequency of functional groups, are presented in Figure 6. As the temperature rises, the stretching vibration absorption of the carbonyl group within 1800–1680 cm^−1^ gradually increases. This indicates that the degradation reaction of POM goes on under heating, and the effects of the carbonyl group enhance this. The broad band of saturated fatty acid (2900–2700 cm^−1^) possibly indicates the existence of formic acid (HCOOH). The significant difference is in the IR spectrum that emerged around 2300 cm^−1^. Under N_2_ atmosphere, the isocyanate group will be generated and reaches its peak at 400 °C. At the temperature range of 150–400 °C, the dehydration reaction will take place in POM, thereby forming the C≡C bond. However, under O_2_ atmosphere, the obvious absorption peak of CO_2_ represents the occurrence of redox reaction, and reaches its peak value at 200 °C, which shows the reaction process is advanced dramatically.

### 4.7. Combustion Performance Analysis

To further investigate the combustion performance of POM, a cone calorimeter was used to perform combustion tests on POM without and with added flame retardants (referred to as NPOM and RPOM), and three radiation intensity values (25 kW/m^2^, 35 kW/m^2^, and 50 kW/m^2^) were set. The data obtained are the results of three repetitions. The ignition times (TTIs) of NPOM and RPOM under the three radiation intensities are shown in Table 5 and Table 6. From the results of the two tables, it can be seen that the TTI of NPOM and RPOM gradually decreases with the increase in thermal radiation intensity, and the decrease of TTI shows a trend of gradual decrease. The TTI data of both samples are further compared in Table 7 and Figure 7. It can be found that the TTI of RPOM is significantly higher than that of NPOM when the thermal radiation intensity is less than 35 kW/m^2^, indicating that RPOM exhibits a less easily ignited performance. However, when the thermal radiation intensity is higher than 35 kW/m^2^, the difference in ignition time between the two is not obvious. Thus, it is assumed that the flame retardant used in this paper does not play a significant role after the thermal radiation intensity exceeds 35 kW/m^2^.

The heat release rate (HRR) curves of NPOM and RPOM under the three heat radiation intensities are shown in Figure 8. As can be seen from Figure 8, the heat release rates of NPOM and RPOM show similar trends: as the heat radiation intensity increases, the peak of HRR increases continuously; meanwhile, the time to reach the peak of heat release rate decreases gradually. For further analysis, the average heat release rate (MHRR), time to peak heat release rate (T_pkHRR_), peak heat release rate (PHRR), and total heat release (THR) parameters of NPOM and RPOM are compared, as shown in Table 8, Table 9, Table 10 and Table 11. It can be seen that RPOM is significantly lower than NPOM in terms of MHRR, PHRR, and THR, indicating that NPOM gives off more heat than RPOM during combustion, and it is presumed that NPOM is more dangerous than RPOM [35]. The difference between the T_pkHRR_ values of the two is minor, especially after the thermal radiation intensity is higher than 35 KW/m^2^; it is presumed that the flame retardant has little effect on the burning rate of POM at this time.

The release smoke rate (RSR) plots of NPOM and RPOM are shown in Figure 9; the relevant data statistics are shown in Table 12. It can be seen that the peak smoke release rate of both NPOM and RPOM increases with the growth of thermal radiation intensity, and the peak smoke release rate of NPOM is significantly higher than that of RPOM in the case of the same thermal radiation intensity. Further analysis of the smoke-release-related parameters of NPOM and RPOM are summarized as shown in Table 13 and Table 14. It can be seen from Table 13 that the total smoke release of NPOM is significantly higher than that of RPOM under the same thermal radiation intensity. It can be seen from Table 14 that there is almost no difference in the time to reach the peak smoke release rate between the two samples, thus it is presumed that the flame retardant can only improve the smoke release parameters of POM and has little effect on the combustion performance parameters, which is consistent with the above inference [36].

## 5. Conclusions

The pyrolytic properties as well as the combustion properties of POM have been investigated with TG–FTIR and combustion tests. The activation energy *E_α_* was calculated with three isoconversional methods (the Friedman, FWO, and KAS methods); it was found that the *E_α_* values are almost constant in the 0.1–0.9 conversion range, showing that thermal degradation for POM is a single-step process. The degradation of POM has been found to obey the “*n* + *m* = 2; *n* = 1.5” mechanism in nitrogen, but the “A3” mechanism in air, as analyzed with the Criado method. The thermostability of POM polymer in air is greatly reduced because the thermal degradation is accompanied by thermo-oxidative degradation, and much better thermal stabilities are revealed for POM samples under inert nitrogen. According to the experimental data of the two types of POM in the conical calorimeter, it was found that the addition of flame retardant effectively reduces the total heat release, thus effectively reducing the fire hazard. This paper can provide references and suggestions for the safe use of POM as well as engineering applications.

## Figures and Tables

**Figure 1 polymers-15-02286-f001:**
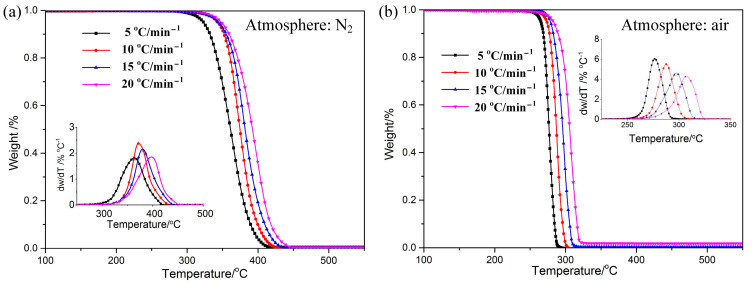
TG−DTG curve s of POM at different heating rates under N_2_ (**a**) and Air (**b**) atmospheres.

**Figure 2 polymers-15-02286-f002:**
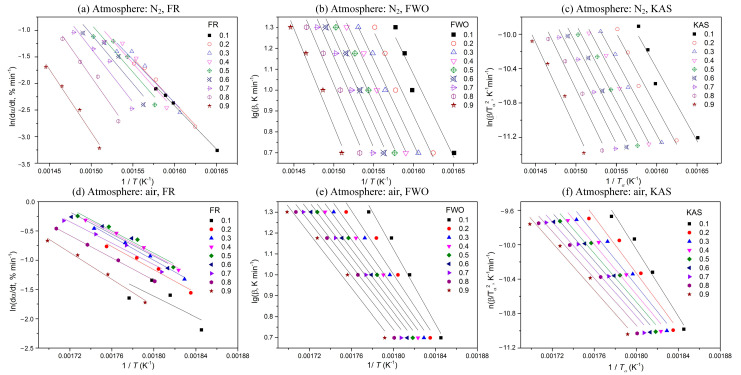
Plots of determination of activation energy at different conversions using different methods.

**Figure 3 polymers-15-02286-f003:**
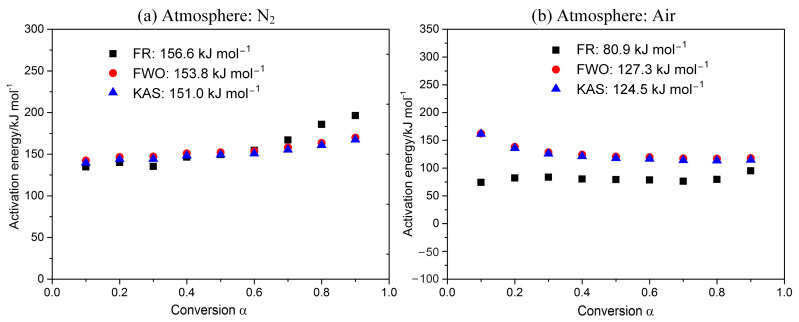
Dependence of activation energy E on the conversion α.

**Figure 4 polymers-15-02286-f004:**
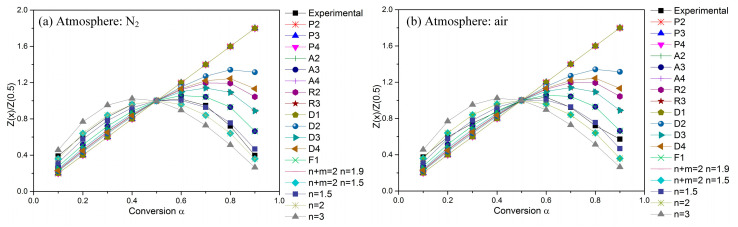
Master curves of different kinetic models and experimental data at 10 °C min^−1^.

**Figure 5 polymers-15-02286-f005:**
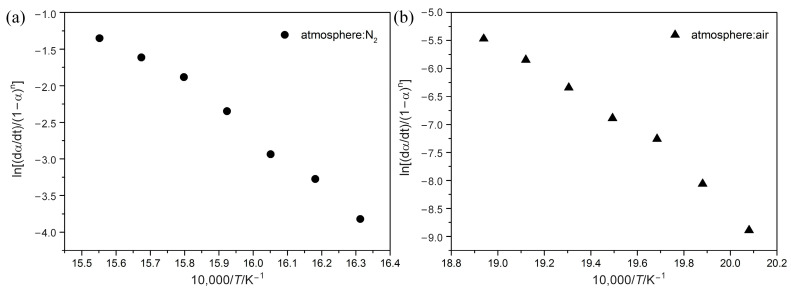
Chang plots of ln[(d*α*/d*t*)/(1-*α*)^n^] vs. 1/*T* at a heating rate of 10 °C min^−1^ under N_2_ (**a**) and Air (**b**) atmospheres.

**Figure 6 polymers-15-02286-f006:**
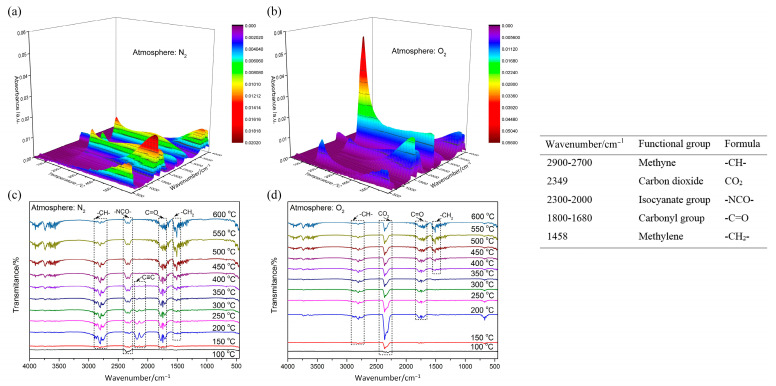
IR spectrum of evolved gas and identified IR frequency of functional groups: (**a**) 3D IR spectrum under N_2_; (**b**) 3D IR spectrum under O_2_; (**c**) 2D IR spectrum under N_2_; (**d**) 2D IR spectrum under O_2_.

**Figure 7 polymers-15-02286-f007:**
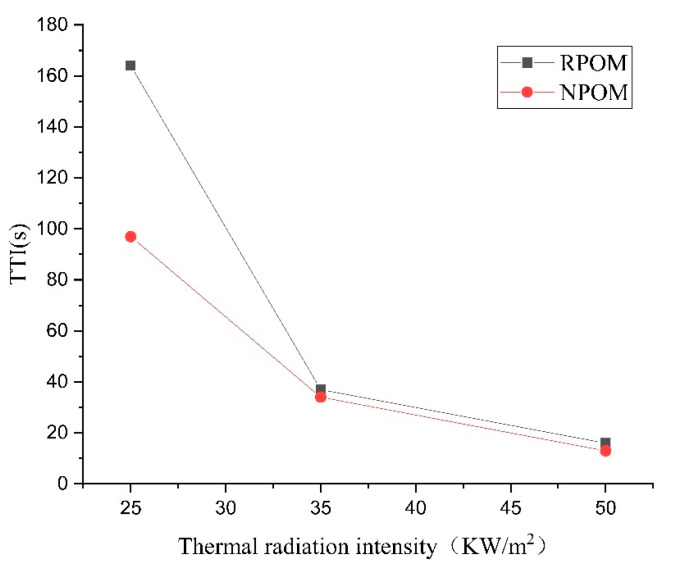
Comparison of thermal radiation intensity–ignition time of NPOM and RPOM.

**Figure 8 polymers-15-02286-f008:**
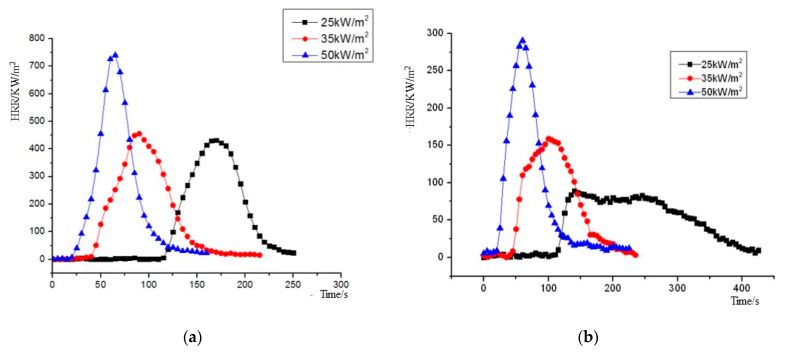
Heat release rate (HRR) plot of NPOM and RPOM. (**a**) NPOM. (**b**) RPOM.

**Figure 9 polymers-15-02286-f009:**
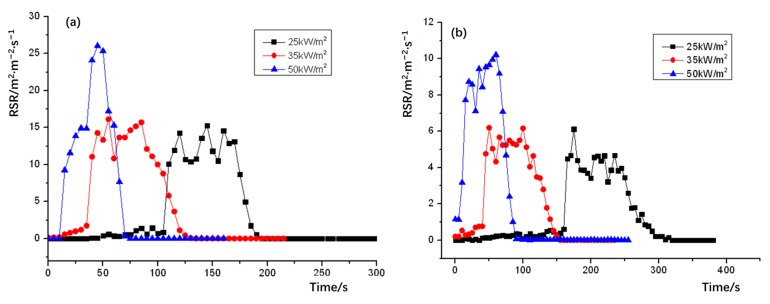
Heat radiation intensity–release smoke rate (RSR) for (**a**) NPOM and (**b**) RPOM.

**Table 1 polymers-15-02286-t001:** Algebraic relationships for *f* (α) and g (α) for kinetic models.

Model	*f* (α)	*g* (α)
P_2_	2α^1/2^	α^1/2^
P_3_	3α^2/3^	α^1/3^
P_4_	4α^3/4^	α^1/4^
A_2_	2(1 − α)[−ln(1 − α)]^1/2^	[−ln(1 − α)]^1/2^
A_3_	3(1 − α)[−ln(1 − α)]^2/3^	[−ln(1 − α)]^1/3^
A_4_	4(1 − α)[−ln(1 − α)]^3/4^	[−ln(1 − α)]^1/4^
R_2_	2(1 − α)^1/2^	[1 − (1 − α)^−1/2^]
R_3_	3(1 − α)^2/3^	[1 − (1 − α)^−1/3^]
D_1_	1/2α	α^2^
D_2_	[−ln(1 − α)]^−1^	[(1 − α)ln(1 − α)] + α
D_3_	3(1 − α)^2/3^[2(1 − (1 − α)^−1/3^]	[1 − (1 − α)^−1/3^]^2^
D_4_	3/2((1 − α)^−1/3^ − 1)	1 − (2α/3) − (1 − α)^2/3^
F_1_	(1 − α)	−ln(1 − α)
*n* + *m* = 2; *n* = 1.9	α^0.1^(1 − α)^1.9^	[(1 − α)α^−1^]^−0.9^(0.9)^−1^
*n* + *m* = 2; *n* = 1.5	α^0.5^(1 − α)^1.5^	[(1 − α)α^−1^]^−0.5^(0.5)^−1^
*n* = 1.5	(1 − α)^1.5^	2[−1 + (1 − α)^−1/2^]
*n* = 2	(1 − α)^2^	−1 + (1 − α)^−1^
*n* = 3	(1 − α)^3^	2^−1^[−1 + (1 − α)^−2^]

**Table 2 polymers-15-02286-t002:** Parameters of mass-loss stages at different heating rates in TG–DTG curves.

Atmospheres	Heating Rates/°C min^−1^	*T*_onset_/°C	*T*_end_/°C	*T*_p_/°C
N_2_	5	333	418	361
	10	354	430	373
	15	359	438	380
	20	365	449	397
Air	5	269	299	276
	10	277	312	288
	15	283	319	297
	20	294	329	307

**Table 3 polymers-15-02286-t003:** Thermal degradation kinetic parameters for POM in nitrogen and air atmosphere according to the Chang model.

Atmosphere	*E*/kJ mol^−1^	*n*	ln*A*/min^−1^	R^2^
N_2_	276.1	3/2	50.4	0.9878
air	244.3	1/3	50.3	0.9841

**Table 4 polymers-15-02286-t004:** Half-life time *t*_1/2_ and estimated lifetime *t*_f_ for the POM.

Temperature/°C	*t*_1/2_ in N_2_/min	*t*_1/2_ in Air/min	*t*_f_ in N_2_/min	*t*_f_ in Air/min
100	4.856 × 10^16^	1.254 × 10^12^	3.045 × 10^15^	1.139 × 10^11^
150	1.314 × 10^12^	1.141 × 10^8^	8.240 × 10^10^	1.036 × 10^7^
200	3.283 × 10^8^	7.417 × 10^4^	2.059 × 10^7^	6.738 × 10^3^
250	4.004 × 10^5^	196.106	2.511 × 10^4^	17.816
300	1.574 × 10^3^	1.460	98.745	0.133
350	15.058	0.024	0.944	0.002
400	0.287	0.001	0.018	0.000

**Table 5 polymers-15-02286-t005:** The ignition time (TTI) of NPOM at different radiation intensities.

Thermal Radiation Intensity (kW/m^2^)	TTI (s)	Average Value (s)
25	108	97
	97	
	86	
35	42	34
	34	
	28	
50	10	13
	13	
	16	

**Table 6 polymers-15-02286-t006:** The TTI of RPOM at different radiation intensities.

Thermal Radiation Intensity kW/m^2^	TTI (s)	Average Value (s)
25	205	164
	127	
	161	
35	38	37
	32	
	40	
50	18	16
	16	
	14	

**Table 7 polymers-15-02286-t007:** Comparison of mean values of TTI of RPOM and NPOM.

Average TTI Value (s)	Thermal Radiation Intensity (kW/m^2^)		
	25	35	50
NPOM	97	34	13
RPOM	164	37	16

**Table 8 polymers-15-02286-t008:** Comparison of MHRR of RPOM and NPOM under three radiation intensities.

MHRR (kW/m^2^)	Thermal Radiation Intensity (kW/m^2^)		
	25	35	50
NPOM	267	278.6	342
RPOM	68.27	93.15	131.26

**Table 9 polymers-15-02286-t009:** Comparison of PHRR–thermal radiation intensity of RPOM and NPOM.

PHRR (kW/m^2^)	Thermal Radiation Intensity (kW/m^2^)		
	25	35	50
NPOM	443.8	467.5	752.3
RPOM	91.2	157.8	293.23

**Table 10 polymers-15-02286-t010:** Thermal radiation intensity–THR of RPOM and NPOM comparison table.

THR (MJ/m^2^)	Thermal Radiation Intensity (kW/m^2^)		
	25	35	50
NPOM	32.27	31.52	33.88
RPOM	19.21	15.42	18.64

**Table 11 polymers-15-02286-t011:** Comparison of thermal radiation intensity–T_pkHRR_ of RPOM and NPOM.

T_pkHRR_ (s)	Thermal Radiation Intensity (kW/m^2^)		
	25	35	50
NPOM	172	92	67
RPOM	142	109	63

**Table 12 polymers-15-02286-t012:** Comparison of peak smoke release rate of RPOM and NPOM.

Peak Smoke Release Rate (m^2^·m^−2^·s^−1^)	Thermal Radiation Intensity (kW/m^2^)		
	25	35	50
NPOM	17.32	19.24	29.21
RPOM	4.29	7.31	12.43

**Table 13 polymers-15-02286-t013:** Comparison of the total smoke release of RPOM and NPOM.

Total Smoke Release (m^2^·m^−2^)	Thermal Radiation Intensity (kW/m^2^)		
	25	35	50
NPOM	927.33	993.22	913.28
RPOM	503.46	483.33	562.14

**Table 14 polymers-15-02286-t014:** Comparison of the time to reach the peak smoke release rate of RPOM and NPOM under three radiation intensities.

Time to Reach the Peak Smoke Rate (s)	Thermal Radiation Intensity (kW/m^2^)		
	25	35	50
NPOM	146	58	44
RPOM	149	56	41

## Data Availability

No applicable.

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
