# Peer review of "Analysis of Thermal Degradation Kinetics of POM under Inert and Oxidizing Atmospheres and Combustion Characteristics with Flame Retardant Effects"

_polymers, 2023, doi:10.3390/polym15102286_

Round 1

Reviewer 1 Report

The article has an interesting topic but I am not satisfied with the presentation of the paper. The article requires a major revision for consideration for publication.

My specific comments are below:

1-Title: Consider modifying the title. It does not sound natural in English and does not include flame retardants 

2-The authors use the word paraformaldehyde but use another abbreviation. Either use PFA or use polyoxymethylene in the text and use POM, please

3-Abstract- Coats Redfern method looks strange and must be removed from the text

4-Why do you use polyformaldehyde in the abstract and in the keywords? please use the same term. Decide the terms paraformaldehyde, polyoxymethylene, or polyformaldehyde.

5- Introduction.

Before application of POM give information about it. What is POM, chemical structure, etc

Explain in the text the function of POM in the applications of mechanical engineering.

6- What is TPU, why didn't you explain it?

7- "Many literatures" is not an acceptable term. You may use "in a number of works" for example. Please give references after it also.

8-Introduction is difficult to understand the objective. Modify the text and explain the objective better. 

9-Also explain the novelty of the work. What makes it different from other articles?

10-Experimental

Write the amount of POM used in thermogravimetry

11-Results and Discussion

Ea of POM is higher by CR method under air? (Table 3).  I do not agree with this. Under air, the energy barrier is lower and it is expected a lower activation energy. Thus, table 3 and the CR method seem to have errors and must be removed from the text

12- The authors did not check the article well before submission. There are spelling mistakes such as "uesd" instead of "used" in the text. Please correct them.

13-The authors used the same values again and again. Please decide to use either a table or a figure, not both.

For example, remove figure 8 or table 8. The same is valid for table 9-fig10, table10-fig 11,table 11-fig 12, table 12-fig 13, table 13-fig 15, table 14-fig 16,table 15-fig 17.

Why didn't you calculate the activation energy with flame retardant use?

14-Conclusions

It is too long. You do not need to repeat what you did in the work. Please only include the key conclusions

I do not agree with "thermostability" explanation. It is possibly about the energy barrier difference.

Remove CR and A3 result from the text

It needs a revision

Author Response

Kindly find the attached response file.

Reviewer 2 Report

The paper presents the investigation of the effect of the ambient atmosphere on the pyrolysis process of paraformaldehyde (POM) using a Thermogravimetric Analyzer and Fourier Transform infrared spectroscopy. However, before consideration for publication, some points must be improved.    

1) Introduction: The authors must include current references in this information “Paraformaldehyde (POM) has been applied increasingly in mechanical engineering, the automotive industry, precision instruments, etc., either alone or as a composite material.”

2) All acronyms that appear in the text for the first time should be specified, for example, TPU.

3) Experimental: What was the mass used for thermal analysis? What are the precision of temperature measurement and the microbalance sensitivity? The authors must include this information in the experimental section.

4) Experimental: Regarding the Fire Testing Technology, did the authors follow any standard test method? The authors should include more details about test conditions.

5) Page 4: The authors wrote: “KAS method [17]” Is this information complete?

6) Page 5, 4.1. Thermal degradation behavior: The authors wrote: “The trend of weightlessness curve is almost the same…” However, according to ICTAC Kinetics Committee recommendations for the analysis of multi-step kinetics (Vyazovkin et al. 2020, Thermochimica Acta, Volume 689, 178597, https://doi.org/10.1016/j.tca.2020.178597.) is more common to use the term mass loss for TGA.

7) In the title of Figure 2, the authors should include the specifications of the acronyms FR, FWO, and KAS to be more didactic.

8) Figure 2: What is the correct atmosphere: air or O2?

9) In Figure 7, the authors should increase the resolution of the plots, especially (a) and (b).

10) I consider the issue relevant to the thermal analysis field, especially for the thermal degradation kinetics and combustion characterization of paraformaldehyde under inert and oxidizing atmospheres. In addition, the paper contributes to the thermal degradation area compared with other published materials.

11) Conclusions are consistent with the arguments presented and address the question proposed. 

The paper presents the investigation of the effect of the ambient atmosphere on the pyrolysis process of paraformaldehyde (POM) using a Thermogravimetric Analyzer and Fourier Transform infrared spectroscopy. However, before consideration for publication, some points must be improved.    

1) Introduction: The authors must include current references in this information “Paraformaldehyde (POM) has been applied increasingly in mechanical engineering, the automotive industry, precision instruments, etc., either alone or as a composite material.”

2) All acronyms that appear in the text for the first time should be specified, for example, TPU.

3) Experimental: What was the mass used for thermal analysis? What are the precision of temperature measurement and the microbalance sensitivity? The authors must include this information in the experimental section.

4) Experimental: Regarding the Fire Testing Technology, did the authors follow any standard test method? The authors should include more details about test conditions.

5) Page 4: The authors wrote: “KAS method [17]” Is this information complete?

6) Page 5, 4.1. Thermal degradation behavior: The authors wrote: “The trend of weightlessness curve is almost the same…” However, according to ICTAC Kinetics Committee recommendations for the analysis of multi-step kinetics (Vyazovkin et al. 2020, Thermochimica Acta, Volume 689, 178597, https://doi.org/10.1016/j.tca.2020.178597.) is more common to use the term mass loss for TGA.

7) In the title of Figure 2, the authors should include the specifications of the acronyms FR, FWO, and KAS to be more didactic.

8) Figure 2: What is the correct atmosphere: air or O2?

9) In Figure 7, the authors should increase the resolution of the plots, especially (a) and (b).

10) I consider the issue relevant to the thermal analysis field, especially for the thermal degradation kinetics and combustion characterization of paraformaldehyde under inert and oxidizing atmospheres. In addition, the paper contributes to the thermal degradation area compared with other published materials.

11) Conclusions are consistent with the arguments presented and address the question proposed. 

Author Response

Kindly find the attached response to reviewer.

Round 2

Reviewer 1 Report

Accepted

Accepted

Reviewer 2 Report

I believe the manuscript has been sufficiently improved to warrant publication in Polymers.